

# Ontogeny of a sexually selected structure in an extant archosaur *Gavialis gangeticus* (Pseudosuchia: Crocodylia) with implications for sexual dimorphism in dinosaurs

David Hone[1], Jordan C. Mallon[2,3], Patrick Hennessey[1,4] and Lawrence M. Witmer[5]

[1] School of Biological and Chemical Sciences, Queen Mary University of London, London, United Kingdom
[2] Beaty Centre for Species Discovery and Palaeobiology Section, Canadian Museum of Nature, Ottawa, ON, Canada
[3] Department of Earth Sciences, Carleton University, Ottawa, ON, Canada
[4] Department of Biology, Georgia Southern University, Statesboro, GA, USA
[5] Department of Biomedical Sciences, Heritage College of Osteopathic Medicine, Ohio University, Athens, OH, USA

Corresponding author
David Hone, dwe_hone@yahoo.com

## ABSTRACT

Despite strong evidence for sexual selection in various display traits and other exaggerated structures in large extinct reptiles, such as dinosaurs, detecting sexual dimorphism in them remains difficult. Their relatively small sample sizes, long growth periods, and difficulties distinguishing the sexes of fossil specimens mean that there are little compelling data on dimorphism in these animals. The extant gharial (*Gavialis gangeticus*) is a large and endangered crocodylian that is sexually dimorphic in size, but males also possesses a sexually selected structure, the ghara, which has an osteological correlate in the presence of a fossa associated with the nares. This makes the species a unique model for potentially assessing dimorphism in fossil lineages, such as dinosaurs and pterosaurs, because it is a large, slow-growing, egg-laying archosaur. Here we assess the dimorphism of *G. gangeticus* across 106 specimens and show that the presence of a narial fossa diagnoses adult male gharials. Males are larger than females, but the level of size dimorphism, and that of other cranial features, is low and difficult to detect without a priori knowledge of the sexes, even with this large dataset. By extension, dimorphism in extinct reptiles is very difficult to detect in the absence of sex specific characters, such as the narial fossa.

## INTRODUCTION

Sexual selection is a major evolutionary driver of many biological traits in animals, and is important for understanding the anatomy, behavior, and evolution of species and clades. One major indicator of sexual selection is sexual dimorphism where one sex is larger than the other and/or shows some form of exaggerated structure absent in the other,

indicating an investment in resources as a means of increasing reproductive success. However, assessing sexual selection is very difficult in extinct groups such as the non-avian dinosaurs (hereafter simply 'dinosaurs') and other reptiles (*Knell et al., 2013*), and attempts to identify sexual dimorphism in dinosaurs have had no real success (see *Mallon, 2017* for a recent review).

Lineages may show only sexual size dimorphism, or dimorphism of major osteological traits (e.g. crests and horns), or may be under mutual sexual selection leading to reduced or even absent dimorphism. Further, dimorphic traits are not necessarily linked to evolutionary pressures based around reproductive success or socio-sexual dominance (see *Hone, Naish & Cuthill, 2012*; *Hone & Mallon, 2017*; and references therein). As a result, the case for sexual selection in dinosaurs and other extinct reptile lineages has been controversial. In some taxa, however, there is evidence for sexual dimorphism (e.g. *Shringasaurus Sengupta, Ezcurra & Bandyopadhyay, 2017*) and for the presence of traits that were likely used as socio-sexual signals (*O'Brien et al., 2018*). Even so, the detection of sexual dimorphism is an important component of understanding sexual selection, and the current lack of evidence for dimorphism in dinosaurs (e.g. *Hone, Naish & Cuthill, 2012*; *Mallon, 2017* and references therein) remains at least a curious anomaly.

There are also limitations to the available models among extant animals for comparison to extinct animals such as dinosaurs. Large mammals may be comparable in size and have some ecological similarities to dinosaurs, but there are major differences in evolutionary histories and growth trajectories. Extant reptiles often show high levels of sexual size dimorphism (*Fitch, 1981*; *Cox, Butler & John-Alder, 2007*) but most reptiles are small and attain adult size rapidly. The extant phylogenetic bracket for dinosaurs consists of birds and crocodylians, making these two groups potentially better candidates (*Witmer, 1995a*). However, as with mammals, birds mature rapidly, and their small size and often limited skeletal trait dimorphism also makes them problematic. Crocodylians, in contrast, may be an excellent model with respect to dimorphism. As with many or even most dinosaurs, they reach large sizes, grow slowly over many years, are sexually mature well before maximum size, lay eggs, and have large numbers of offspring. Importantly, at least some show sexual dimorphism in body size (*Caiman*—*Thorbjarnarson, 1994*; *Alligator*—*Wilkinson & Rhodes, 1997*; *Crocodylus*—*Platt et al., 2009*).

Among extant crocodylians, the gharial (*Gavialis gangeticus*) is a uniquely appropriate example (Fig. 1). Gharials are specialised piscivores having an unusually long and slender snout (*Whitaker & Basu, 1983*). Now known in the wild only from India, Bangladesh, and Nepal, this species is critically endangered, with the already small wild populations having suffered significant losses in recent decades (*Hasan & Alam, 2016*; *Lang, Chowfin & Ross, 2019*). They are among the largest of the extant crocodylians, with the largest recorded animal (a male) reaching 6.25 m in total length and weighing 977 kg, although more typical adults are 3.5–4.5 m long, with males being larger than females (*Hasan & Alam, 2016*).

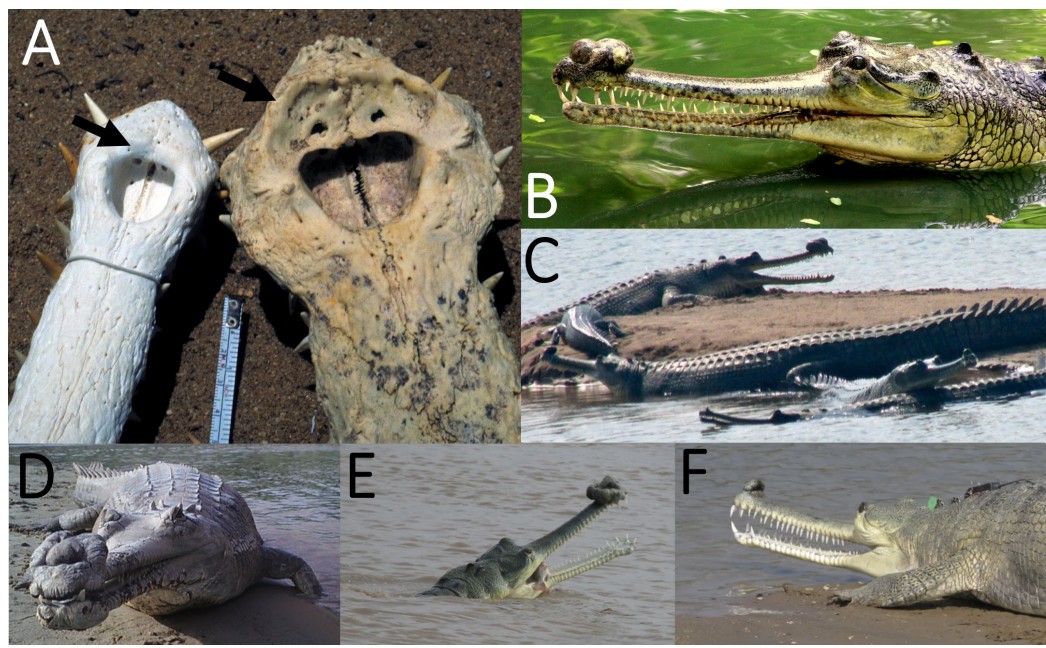

**Figure 1 Gharial snouts and the ghara.** (A) Bony snout of a female (left) and male (right) *Gavialis gangeticus*. The black arrows point to equivalent areas of the skull; note the male's large narial fossae to which the ghara attaches. (B–F) various male *G. gangeticus* in the wild showing the range of size and morphology of the ghara. Image (B) provided by Nikhil Whitaker and images (C–F) provided by the Gharial Ecology Project, all used with permission.  

Gharials show sexual dimorphism not just in body size, but also in their cranial anatomy. Larger males bear a ghara—a growth on the rostrum that is absent in females (*Martin & Bellairs, 1977*; *Biswas, Acharjyo & Mohapatra, 1978*; *Whitaker & Whitaker, 1989*). The ghara is a soft-tissue structure that grows behind and over the external nares, and is supported by a depression on the bony rostrum, anterior to the nares (*Iordansky, 1973*), in the larger male skulls (*Martin & Bellairs, 1977*). Some early descriptions of the ghara suggest that it is bony and may even be inflated, but this is not the case (*Martin & Bellairs, 1977*).

The exact function of the ghara is uncertain, but as it only appears in larger and presumably sexually mature males (*Martin & Bellairs, 1977*; *Biswas, Acharjyo & Mohapatra, 1978*; *Whitaker & Whitaker, 1989*), it would be reasonable to assume that it functions in sexual display. Large males are seen to be dominant over smaller males and females (*Whitaker & Basu, 1983*). Suggested functions of the ghara include altering the calls of males (a hissing sound not made by females or young males; *Whitaker & Whitaker, 1989*), or as a visual display signal to females (*Martin & Bellairs, 1977*). Large males also possess an additional secondary sexual characteristic consisting of a pair of expanded bony bullae on the dorsal aspect of the pterygoid bones (Fig. 2), as also seen in other crocodylians (e.g. *Alligator*—Fig. 3). In gharials, the bullae are not present in small males or apparently in females (though this is uncertain). The egg-shaped bullae are dorsal dilations of the bony nasopharyngeal duct (*Wegner, 1958*; *Witmer, 1995b*, *1999*). Although the resonant properties of the pterygoid bullae have yet to be demonstrated

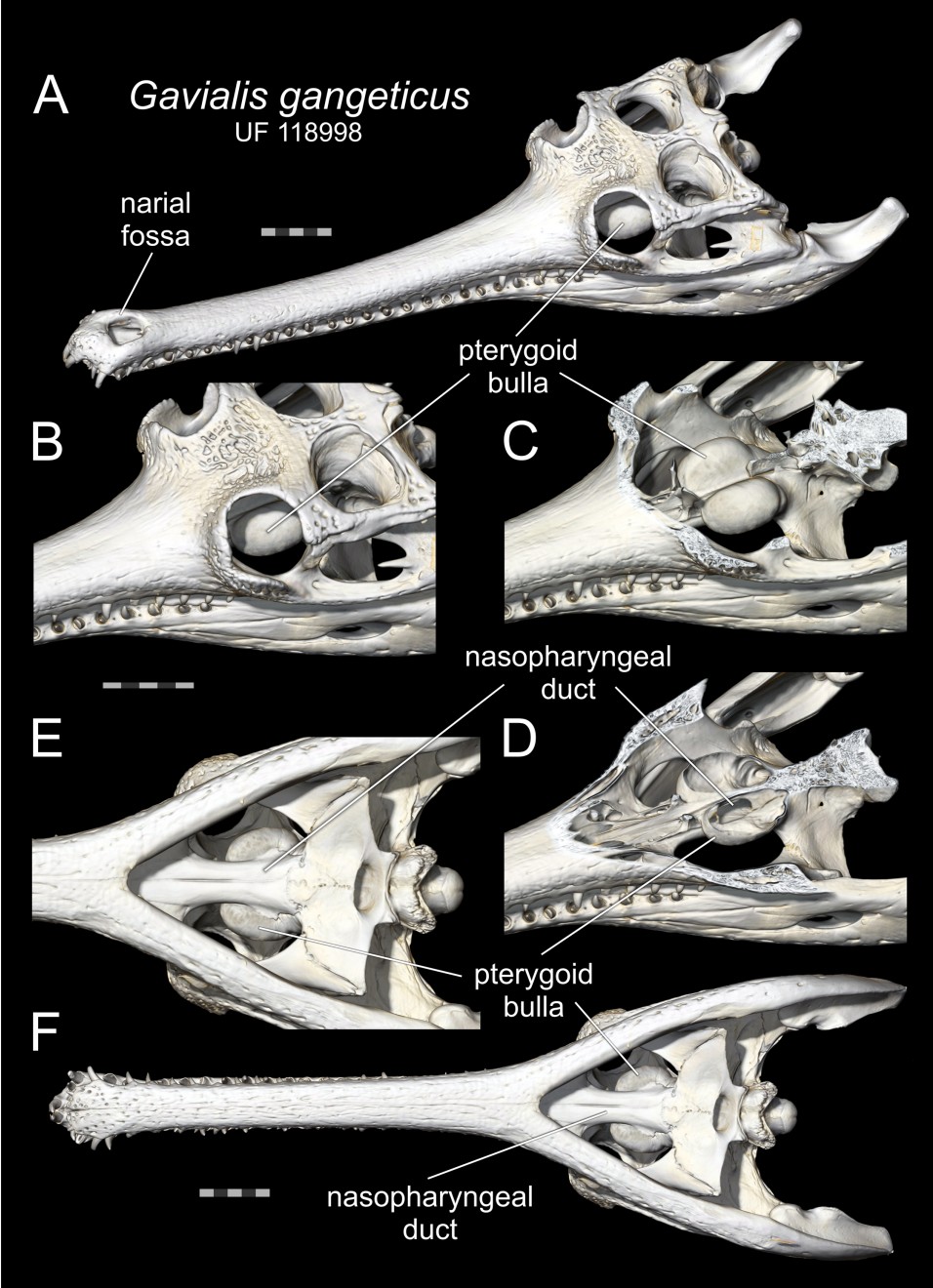

**Figure 2 Pterygoid bullae of gharials.** Pterygoid bullae of *Gavialis gangeticus* (UF 118998) based on volume renders of computed tomographic data of a dried skull. (A) Dorsolateral oblique view of the full skull showing the immature narial fossa and the pterygoid bulla, the latter being seen through the orbit. (B–E) Dorsolateral oblique views of (B) the pterygoid bulla enlarged, (C) with the skull roof digitally removed to reveal both bullae, and (D) with the dorsal portions of the bullae removed to show the thin walls and the connection with the nasopharyngeal duct. (E) Close-up ventral view of the pterygoid bullae projecting into the adductor muscle chamber. (F) Ventral view of the full skull. All scale bars equal 5 cm. (A) and (F) are at the same scale, as are (B–D).

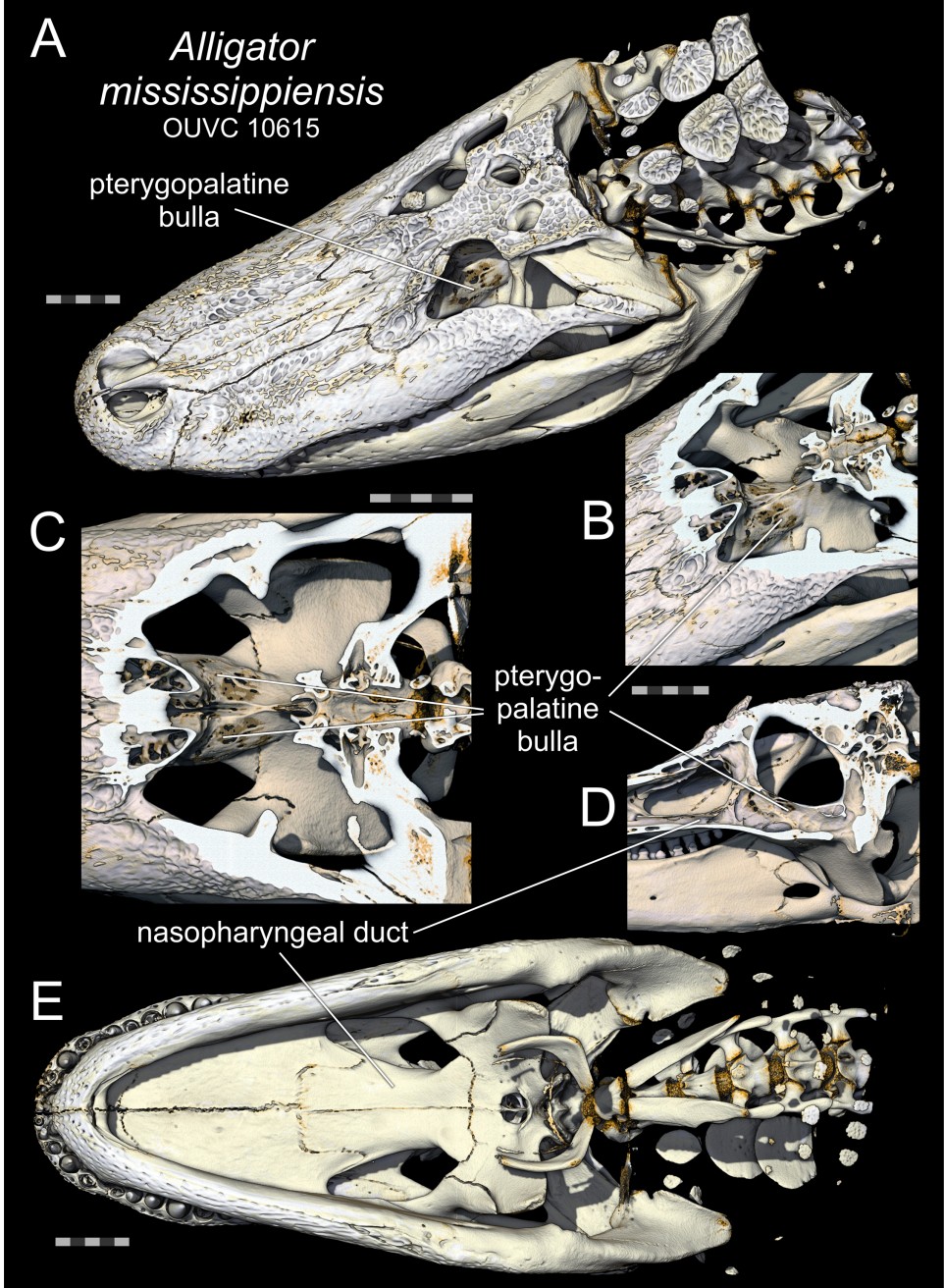

**Figure 3 Pterygopalatine bullae of *Alligator*.** Pterygopalatine bullae of *Alligator mississippiensis* (OUVC 10615) based on volume renders of computed tomographic data of a fleshy head. (A) Dorso-lateral oblique view of the full skull showing the pterygopalatine bulla as seen through the orbit. (B) Dorsolateral oblique view of the pterygopalatine bulla enlarged with the skull roof digitally removed to reveal both bullae. (C) Same presentation as in (B) but in dorsal view and enlarged. (D) Medial (internal) view of the right side of a parasagittaly sectioned head, showing the pterygopalatine bulla emerging as a dorsal dilation of the nasopharyngeal duct. (E) Ventral view of the full skull showing that the pterygopalatine bulla is not visible in ventral view. All scale bars equal 5 cm. (A) and (E) are at the same scale, as are (B) and (D), whereas (C) has its own scale.

experimentally, they would function as vocal resonating chambers as an inescapable bioacoustic or biophysical consequence of their connection to the vocal tract. Thus, bullae may be linked to the ghara, functioning in sound production or modification (*Martin & Bellairs, 1977*). However, vocalisations are rare (*Whitaker & Basu, 1983*), potentially arguing against this interpretation.

Ontogenetic data are limited to a few scattered accounts. For example *Biswas, Acharjyo & Mohapatra (1978)* describe a male of c. 2.5 m total length, and aged 11.5 years, as showing the first signs of a ghara. The same animal was described as showing 'sex play' (which we assume to mean courtship behaviour), aged 12.5 years, suggesting that the growth of the ghara is linked to maturity (females appear to mature at around 2.6 m in length; *Whitaker & Basu, 1983*). Similarly, *Whitaker & Whitaker (1989)* described one male as having a snout resembling that of a female until it was 11 years old, when the ghara started to develop; the ghara folded caudally over the nostrils at age 14 years, and reached fully adult form by age 18 years. *Martin & Bellairs (1977)* suggested that males around 3 m long will exhibit a small ghara, though they also referred to previous reports suggesting that this is normally present only in males in excess of 4.5 m long. Clearly, however, this is a feature that is not present in small/young animals. *Moore et al. (2019)* observed that onset of puberty in male Morelet's crocodiles (*Crocodylus moreletii*), measured by the development of the penile glans, was coincident with changing cranial shape, suggesting a potentially similar pattern.

Here we look at sexual dimorphism in the skull of *Gavialis* as a model for detecting sexual dimorphism and the identity of specimens in extinct reptiles, including dinosaurs. We use the largest known sample of gharial data to assess sexual dimorphism in these animals and to examine the feasibility of detecting dimorphism in extinct reptile lineages. This is done by examining the relative growth (allometry) of changes in various measurements and features of the skull, including the fossa and pterygoid bullae, and searching for evidence of bimodality among adult individuals.

## MATERIALS AND METHODS

Two binary variables and 13 continuous variables (Fig. 4; Appendix 1) were collected from 106 gharial skulls accessioned in 36 museum collections around the world. Where possible, these were measured first-hand with callipers, but it was necessary to measure most of them digitally based on photographs including scale bars. It was impossible to measure all variables because some skulls were incomplete or variably covered with skin. Sex data were not given for any specimens. We therefore assumed that specimens bearing a narial fossa (the osteological correlate of the ghara) were male; those lacking a narial fossa were assumed to be immature and/or female. The narial fossa (Figs. 1, 2 and 4) is the depression in the premaxillary bones adjacent to the bony nasal aperture (i.e. the opening into the nasal passage). A small ghara has been reported in a captive animal that, when dissected, was seen to have ovaries, but this was assumed to have been a pathological individual (*Martin & Bellairs, 1977*). It is therefore reasonable to assume that animals with a narial fossa are male, and hereafter all mentions of male specimens are based on this assumption.

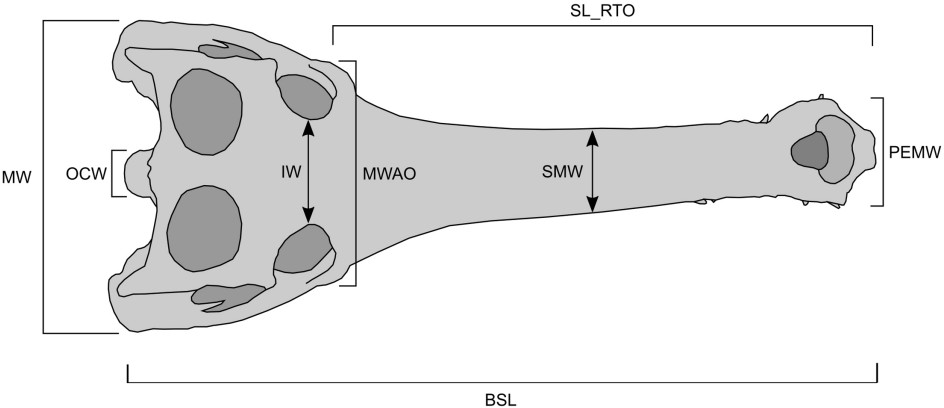

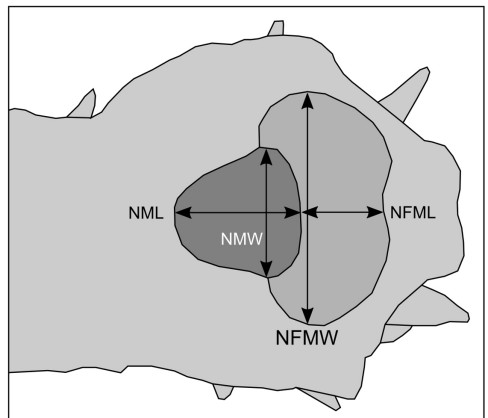
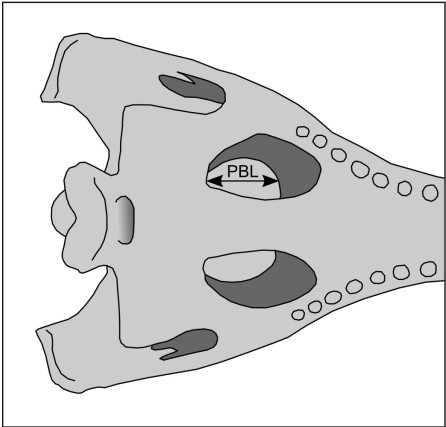

**Figure 4 Measurements used in the present study.** Left inset details measurements from the rostrum; right inset details measurement from the palate. See main text for the key to the abbreviations.

To assess allometry in those continuous variables, it was necessary to first choose a regressor. Visual inspection of the skulls, and prior published work (*Piras et al., 2014*), suggested that variables associated with the snout vary widely in the adults, so we chose maximum skull width (MW), measured across the outside of the quadratojugals, as our regressor, which is consistent with the recommendations of *O'Brien et al. (2019)*. The data were initially log-transformed then subjected to reduced major axis (RMA) regression, which accounts for measurement error in both the independent and dependent variables. Isometry was rejected if the confidence intervals of the regression slope did not bound a value of 1. Negative allometry was manifest if the confidence intervals were <1; positive allometry was manifest if the confidence intervals were >1.

We used the gharial data to attempt to model the detection of sexual dimorphism in the fossil record by disregarding sex information (inferred from narial fossa presence/absence). As advocated by *Mallon (2017)*, we tested for dimorphism in the continuous data by first subjecting the residuals of the RMA regressions to Shapiro-Wilk and Anderson-Darling tests for normality ($\alpha = 0.05$). Residuals were then subjected to a supplemental Hartigan's dip test, which yields the likelihood that the data are distributed unimodally (*Hartigan & Hartigan, 1985*). Residuals were also subjected to mixture

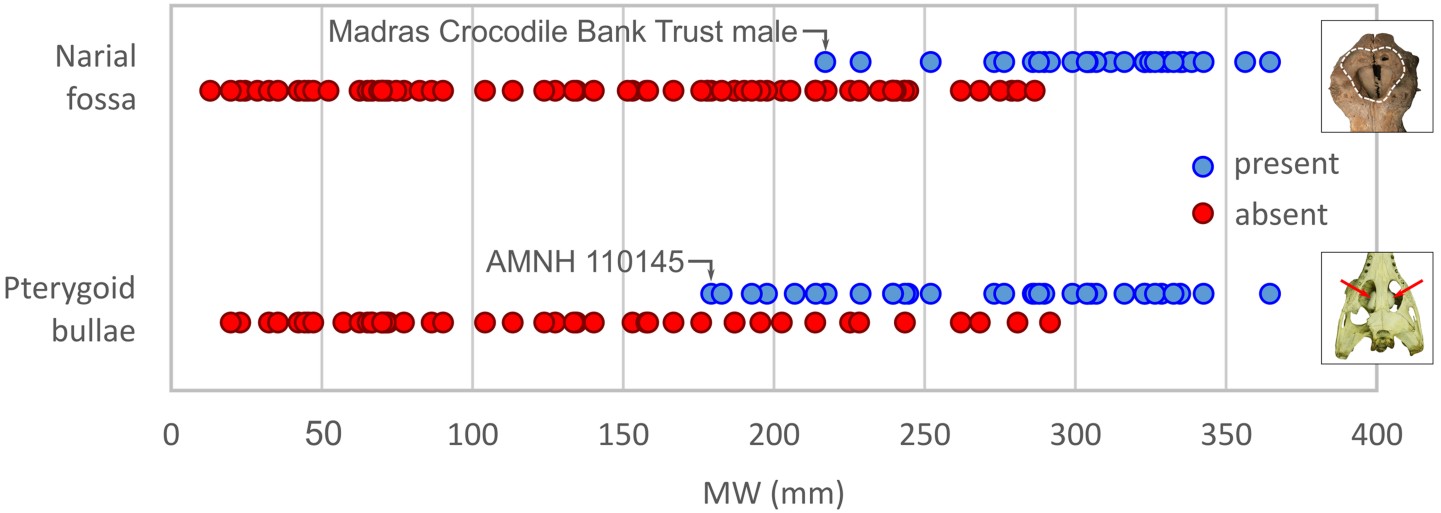

**Figure 5 Gharial narial fossa and pterygoid bullae presence/absence.** Distribution of narial fossa and pterygoid bullae presence/absence across gharial skulls of different sizes (measured as MW). Insets at right illustrate the variable in question.

analysis, a maximum-likelihood procedure used to estimate univariate parameters (e.g. mean, standard deviation) of two or more univariate normal distributions from a pooled sample. The fit of the models was assessed using the Akaike Information Criterion (AIC).

As a final attempt to quantify sexual dimorphism, we reasoned that dimorphic structures should exhibit higher variance of the RMA residuals than non-dimorphic structures. To test for this, we used Levine's test for homogeneity of variance from means, with follow-up $F$-test pairwise comparisons. The multiple comparisons were adjusted using Holm–Šidák correction.

## RESULTS

### General observations

Maximum skull width (MW) ranges between 13 mm and 356 mm (embryonic and large adult skulls, respectively) in our dataset, with basal skull lengths (BSL) ranging from 33 mm to 864 mm, and total estimated lengths of the animals between 17 cm and 5.9 m (see Appendix 1). More than 30 of our specimens represent animals of estimated total length in excess of 5 m, showing that this dataset is biased towards larger animals, presumably at least in part as a result of selective acquisition of large specimens for museums. The smallest skull having a narial fossa is the Madras Crocodile Bank Trust male, where MW = 217 mm and BSL = 581 mm (Fig. 5) or approximately 60% maximum size. Above MW = 280 mm (BSL c. 715 mm), all skulls possess a narial fossa; thus, the largest skulls are male. The smallest skull possessing pterygoid bullae (but lacking a narial fossa) is from the American Museum of Natural History (AMNH) 110145, where MW = 179 mm (BSL = 459 mm).

Thirty-one of the 106 skulls possess narial fossae. Where determinable (some lack palates or are otherwise obscured), all of these possess pterygoid bullae, indicating

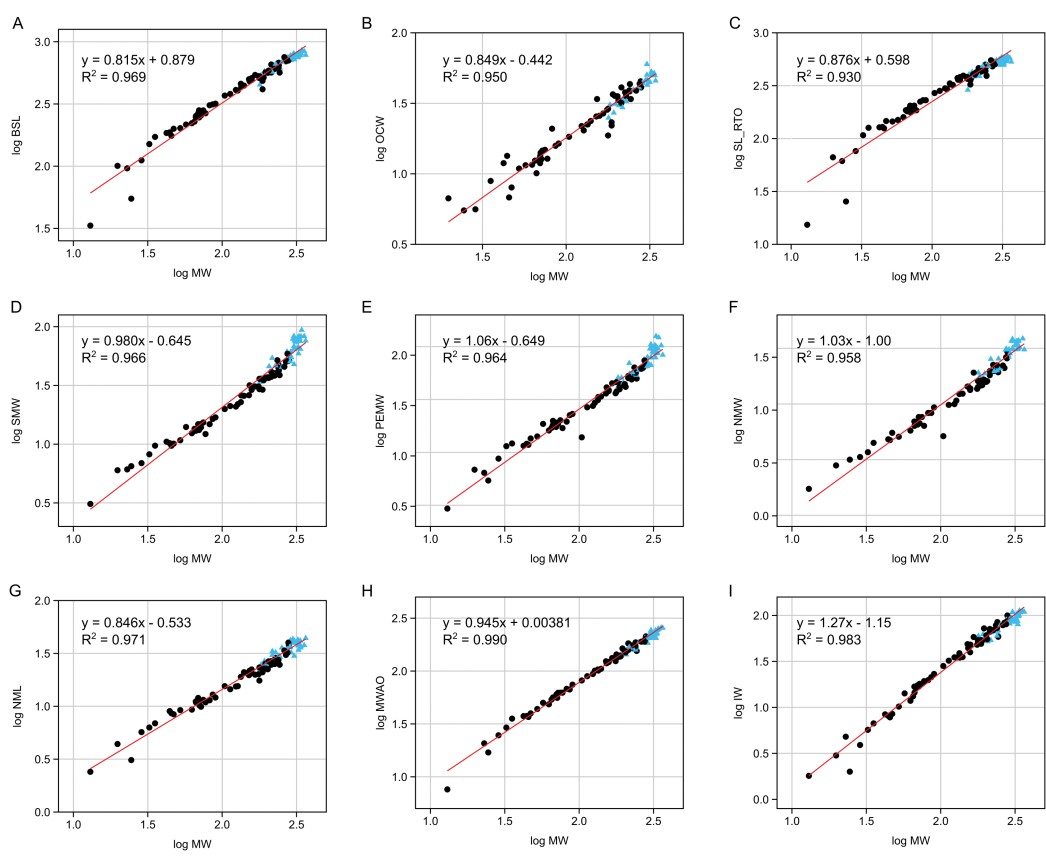

**Figure 6 Reduced major axis regressions for nine of the assessed continuous variables.** (A) Log BSL vs. log MW, (B) log OCW vs. log MW, (C) log SL_RTO vs. log MW, (D) log SMW vs. log MW, (E) log PEMW vs. log MW, (F) log NMW vs. log MW, (G) log NML vs. log MW, (H) log MWAO vs. log MW, (I) log IW vs. log MW. Males (possessing narial fossae) are identified with blue triangles. Black dots represent osteologically immature individuals and/or females.

that they are predominantly, if not universally, present in males. There are, however, 11 skulls lacking a narial fossa but having bullae (Fig. 5). Of these, six are smaller than the unnumbered Madras Crocodile Bank Trust male, and all are smaller than MW = 280 mm (BSL = 743 mm), which is the lower threshold at which the narial fossa is consistently expressed (Fig. 5). If having a pterygoid bulla is indicative of the male sex, which seems likely, it shows up earlier in ontogeny, at approximately 50% maximum size, than does the narial fossa.

## Allometry

The results for the all-inclusive allometric analyses are summarized in Fig. 6 and Table S1. MW is a good predictor of all continuous cranial variables ($R^2 > 0.92$, $p < 0.0001$). Positively allometric variables include PEMW (premaxillary expansion maximum width) and IW (interorbital width). Negatively allometric variables include BSL (basal skull length), OCW (occipital condyle width), NML (naris maximum length) and MWAO (maximum width across orbits). The three remaining variables—SL_RTO (snout

length rostral to orbits), SMW (snout minimum width) and NMW (naris maximum width)—grow isometrically.

The largest of our presumed males consistently plot above the regression line for SMW, PEMW and NMW (Fig. 6), suggesting that these variables increase very rapidly at large body sizes. Follow-up allometric analyses of just the male data (having pterygoid bullae and narial fossae) reveal positive allometry in all three variables (Table S2), with slopes greater than those reported for the entire dataset, albeit with slightly reduced goodness of fit ($R^2 > 0.75$, $p < 0.0001$).

Narial fossa maximum length and narial fossa maximum width (NFML and NFMW, respectively) scale with positive allometry, but the relationship with MW is insignificant (Table S2). PBL (average pterygoid bulla length) is weakly but positively correlated with skull size in males. We were unable to reject the null hypothesis of isometry (Table S2).

### Sexual dimorphism

Presumed males categorically differ from females in the presence of a narial fossa (~ghara) and, ostensibly, the presence of pterygoid bullae. Males are further distinguished by their absolutely larger skulls at maturity (MW > 287 mm, BSL > 715 mm), relatively shorter and wider rostra, and wider terminal rosettes that support larger nares.

Without knowing the sexes a priori (as in fossil taxa), it is otherwise difficult to detect dimorphism in those continuous variables. With the exception of NML, the residuals for all cranial variables are significantly non-normal, but in no case are they significantly non-unimodal (Table S3). Mixture analysis shows that pooled BSL values are best modelled by two normal distributions, but further investigation shows that these distributions distinguish between juvenile and more mature individuals, not sexual dimorphs (Group 1 mean = −0.205 ± 0.0739, Group 2 mean = 0.00617 ± 0.0312). The ability to detect dimorphism in the three most obviously dimorphic continuous variables (SMW, PEMW and NMW) does not increase by considering only adults (MW ≥ 179 mm, BSL ≥ 459 mm the smallest presumed male having pterygoid bullae). The adult NMW residuals are not significantly non-normal, and none of the residuals are either significantly non-unimodal or best fit by two normal distributions (Table S3).

On average, the SMW, PEMW and NMW residuals exhibit higher variances (≥0.004) than the non-dimorphic residuals (Fig. 7). Levene's test for homogeneity of variance from means is highly significant ($p < 0.0001$). Follow-up pairwise comparisons (Table S4) reveal that variance for the SMW, PEMW and NMW residuals is usually significantly higher compared to the other residuals. Variance does not differ significantly between any of the three dimorphic variables.

## DISCUSSION

### Sexual dimorphism and sexual selection in *Gavialis*

The results here broadly align with previous assessments of dimorphism and the ghara in *G. gangeticus*. Males are larger than females, and the former show both a fossa associated with the nares and bullae on the palate. These latter features appear only in larger animals,

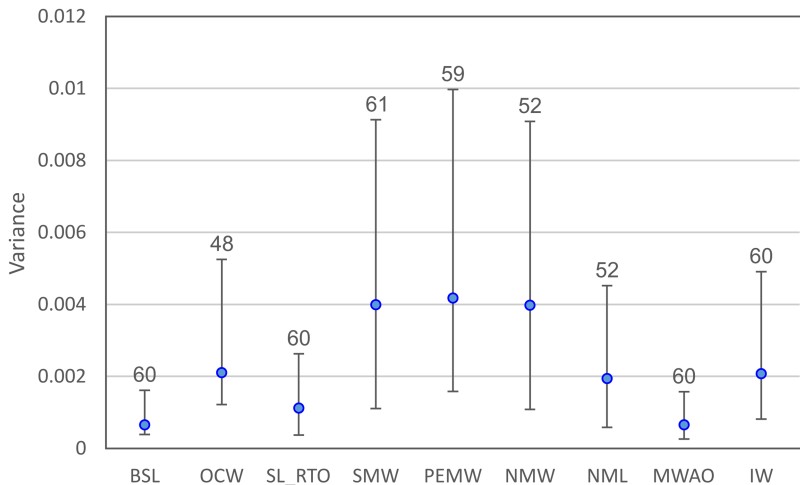

**Figure 7 Residual variances (±95% confidence intervals) for the nine continuous variables examined.** Numbers provided above each of the variables indicate sample size.

and thus with the onset of sexual maturity. The smallest specimen with a narial fossa in our sample is slightly smaller, at 581 mm in basal skull length (MW = 217 mm), than the smallest reported by *Martin & Bellairs (1977*; 690 mm*)*, but the samples are broadly comparable. There are several large animals (with BSL > 700 mm, MW > 270 mm) that lack a fossa, and these would be considered females, though they would be unusually large with body sizes over 4.5 m.

*Martin & Bellairs (1977)* stated that a ghara appears in males of around 3 m in total length, which, following their head to body ratio of 1:6, would equate to a basal skull length of approximately 420 mm. This is considerably smaller than the smallest skull with a narial fossa recorded here. We infer from this finding that the ghara may be growing for some time before the ghara's soft tissues are extensive enough to generate the bony fossa that supports it (just as muscle scars tend to be subtle in small animals and more prominent in large animals). This would match with the apparent development of the bullae prior to the presence of the fossa and would indicate that the ghara and bulla develop nearly simultaneously. Above a BSL of 680 mm (MW > 240 mm), there are 23 specimens where the presence of a fossa and bullae can be reliably determined (see Appendix 1). Of these, 20 possess both, two possess neither, but one specimen (Royal Scottish Museum, RSM 1948.58.1) has bullae and no fossa. Thus, there is not a perfect 1:1 ratio between larger animals possessing both a fossa and bullae or neither, though clearly a strong majority do fall into this category. We therefore provisionally consider the bullae to be a feature of adult male animals, but recognise that this may not be exclusive.

The narial fossa shows strong positive allometry compared to other traits in the skull (Fig. 6), which would suggest that the affixed ghara functions as a sexually selected display feature (cf. *O'Brien et al., 2018*). This also fits with the observation of *Dinets (2013)* that males have a sex-specific head-up posture on land, which elevates the ghara. The ghara

would also be a major handicap to males when hunting, and thus would form an honest signal of the fitness of the owner. The large size of the ghara (Fig. 1) might also increase visibility to prey, and would certainly offer considerable drag on the otherwise thin snout of an animal hunting in water (*Martin & Bellairs, 1977*), presumably incurring a cost to feeding effectiveness. This is especially true since an extra drag generated near the tip of the jaws would be much greater than closer to the rear of the skull, as drag is a function of distance from the joint squared. The extreme variation of the morphology of the ghara (Fig. 1) also points to it being a socio-sexual signal (see *Hone & Naish, 2013* and references therein) that would be under selection.

Initially, the pterygoid bullae must grow rapidly as the smallest record of them in our sample is still sizeable (37 mm long on the Field Museum of Natural History (FMNH) 22025, BSL = 611 mm, MW = 218 mm), although it is possible that at smaller sizes they are hidden in photographs of the palate. However, bulla growth is isometric, which suggests that, although they are important structures, their size is not critical. We hypothesise, therefore, that these features function as an acoustic signal to females (or perhaps other males) that the male is mature, but that there is no additional information about the size and quality of the male possessing them or otherwise their growth would be expected to be positively allometric.

Although it has been noted that gharials rarely produce calls, they are known to vocalise (*Whitaker & Basu, 1983*), including 'buzzing' during courtship (*Dinets, 2013*). Many crocodylians communicate using very low-frequency vocalizations (*Garrick, Lang & Herzog, 1978*), some of which extend into the infrasonic range (i.e. below the range of normal human hearing; *Todd, 2007*). Although gharials are unusual among the larger crocodylians, in that they are not known to produce infrasonic calls (*Dinets, 2013*), these or other vocalisations may not have been detected by previous researchers. The bullae are connected to the vocal/respiratory tract and would act as acoustic resonators, potentially lowering the frequencies of sounds produced. Whether the pterygoid bullae are important for acoustic signalling remains unconfirmed. They are, however, large structures that we presume have some positive function, in that they occupy space in the orbit and palatal regions, and would presumably adversely affect other functions (e.g. the bullae expand into the adductor chamber and hence decrease the available space for jaw adductor muscles such as *M. pterygoideus dorsalis*). They appear in larger, mature males, and infrasonic calls of other crocodylians are produced only by males (*Dinets, 2013*). The vocal capabilities of alligators are well known (*Garrick, Lang & Herzog, 1978*; *Vliet, 1989*; *Todd, 2007*), and alligators also possess inflations of the nasopharyngeal ducts known as the pterygopalatine bullae (*Wegner, 1958*; *Witmer, 1995b*, *1999*; though these are presumably non-homologous), perhaps lending some credence to an acoustic resonance function in gharials, as well (Fig. 3). *Dinets (2013)* reported specifically that gharials do not use infrasound, but the basis for this assertion is not clear, and we regard acoustic signalling (potentially including an emphasis on low-frequency sounds, perhaps even infrasound) as the current best-supported hypothesis for the function of the pterygoid bullae.

When analysed in the presumed males alone, some of the traits in the skull also show positive allometry. The premaxillary expansion maximum width (PEMW), the snout

minimum width (SMW) and the naris maximum width (NMW) are all disproportionately larger in the largest males (Fig. 7). All are potentially associated with the ghara and the size increase of the fossa to which it attaches. We suggest here that the expansion of the premaxillary rosette and relative increase of the snout minimum width could help strengthen the skull given the drag of a large ghara, but could also potentially have an ecological function permitting engaging of larger prey.

Given that the pterygoid bullae ontogenetically appear in presumed males that would be of breeding age, we suggest that they provide a general infrasonic acoustic signal that would function to advertise their maturity to females (and also perhaps to other males). This signal would serve to attract attention to the male (even while out of line of sight, such as underwater), and the primary visual signal of status and quality would be the ghara. The visual signal is enhanced by a spray of water emerging from the ghara itself upon exhalation, accompanied by an audible hiss and hum (*Whitaker & Whitaker, 1989*).

## Detecting sexual dimorphism in the fossil record

To date, no dinosaur has been determined to exhibit sexual dimorphism under rigorous analysis (*Mallon, 2017*). Tests for dimorphism in fossil taxa may be confounded by a combination of small sample sizes and protracted growth, coupled with uncertainty of the age or sex of most specimens. These conspire to ensure that young individuals of the larger sex are conflated with older individuals of the smaller sex (*Hone & Mallon, 2017*). Dinosaurs matured sexually before they reached growth asymptotes (e.g. *Erickson et al., 2007*; *Lee & Werning, 2008*), and as a result may be expected to have initiated the growth of a sexually selected structure earlier in ontogeny than in animals where they reach maximum size with sexual maturity. This also fits with the high juvenile mortality of dinosaurs, and thus may have promoted early reproduction (*Hone & Mallon, 2017*). Thus, although we may expect to, and do, see strong positive allometry for features such as crests and horns under sexual selection (*Hone, Wood & Knell, 2016*; *Brown, 2017*), these features may start earlier and grow more slowly than in traditional models, such as large mammals.

Our results support these general contentions that dimorphism is very difficult to detect in taxa showing growth over considerable periods of time. Were these animals recovered from the fossil record, the presence of the fossa and bullae give clear osteological characters that do not have an obvious mechanical function (sensu *Hone, Naish & Cuthill, 2012*) and appear only in larger specimens, and these would likely be regarded as indicative of sex. *Gavialis gangeticus* is identified by numerous osteological traits (*Iordansky, 1973*) present in all specimens that would signify all specimens as belonging to a single species. However, in the absence of these discrete traits, determining dimorphism would be very difficult. *Hone & Mallon (2017)* assessed detection of sexual dimorphism in alligators based on body size, and suggested a minimum of 60 animals might be needed to statistically support dimorphism, even when the difference between sexes was strong and could be measured effectively. Here in the gharials, there is no clear statistical signal for any continuous traits producing two clusters across all specimens, despite a dataset of over 100 specimens. Even when our approach is restricted to osteological mature
individuals, the signal is weak and arguably present only in some features that are also associated with the ghara. Were these fossilized gharial specimens, and having no a priori knowledge of their dimorphism, there would be little to separate out the sexes.

There are few dinosaur datasets in excess of even 50 specimens where traits such as size and potentially dimorphic display features can be reliably measured. The sample sizes and levels of specimen completeness in most assemblages are low enough that dimorphism will be difficult to detect unless there are clear presence/absence traits or very different morphologies between sexes. This is not say that sexual dimorphism was not present in dinosaurs—and more detailed analyses of assemblages with large samples and completeness (e.g. *Coelophysis*; *Griffin, 2018*) are merited—only that it is difficult to demonstrate quantitatively.

## Future work and conservation implications

Further work is needed to confirm the hypotheses laid out here. An exact relationship between the timing of sexual maturity and the physical expression of the ghara, narial fossa and pterygoid bullae is key to understanding gharial reproductive biology. A formal assessment of any social or mating displays, and the differing acoustic and visual components of this, are also important and may provide information critical to breeding efforts given the severe extinction risk of this species (*Lang, Chowfin & Ross, 2019*).

This study also raises additional issues regarding the functional morphology of gharials, which may also be important for understanding their ecology and behaviour. The ghara will induce severe drag on the jaws during prey apprehension underwater, while the bullae will affect the palatal muscles, which will influence the functioning of the jaws.

Finally, we note that the largest narial fossae are associated with an increase in the size of the terminal rosette and a broadening of the snout (increased minimum width), and this may increase the ability of males to catch larger prey. Although we did not measure tooth size across all specimens, we note that some of the largest males (e.g. Grant Museum—LDUCZ X215) apparently have disproportionately large teeth compared to smaller animals, and this would likely allow them to tackle larger prey than may be expected. Given the great differences in size between osteologically mature gharials and young juveniles, there would be niche partitioning between various different growth stages, as seen in other crocodylians (*Dodson, 1975*). However, there may also be separation between larger (and presumably fitter) gharial males and other adult animals, and if so, this may also be a very important consideration for sustaining populations. If high-quality males are removed from a population, this has the potential for profoundly negative effects in small populations (*Knell & Martínez-Ruiz, 2017*). So if our hypothesis about prey preference is correct, suitable prey for larger males must be a consideration in establishing suitable habitats for sustainable gharial populations or they may be at risk.

*Moore et al. (2019)* noted an increase in size in the anterior teeth of male Morelet's crocodiles (*Crocodylus moreletii*) at sexual maturity, and suggested that this increase was linked to male-male interactions over females. This could also be the case in gharials. However, developmentally this change occurs much earlier in the crocodile than the shift suggested here, and it was only the anterior teeth that changed in the crocodile and

not the entire tooth row, as we observed in *Gavialis*. This difference implies that these respective changes in tooth morphology are not synonymous. In any case, as noted by *Moore et al. (2019)*, increased head size is correlated with increased bite power and opportunities to tackle larger prey (*Erickson et al., 2012*). So regardless of the selective pressures that might produce more robust crania and teeth in large males, the hypothesised dietary shift may still be present.

## CONCLUSIONS

We agree with previous studies that the ghara and associated narial fossa, and probably the pterygoid bullae, are male traits of gharials, and most likely have a socio-sexual function in displays. Limited dimorphism in size and various cranial traits are exhibited with males being larger than females.

In the absence of key traits, demonstrating moderate sexual dimorphism (body size or other measurements) is extremely difficult, even with good sample sizes and complete sets of data. Doing so for such fossil taxa as dinosaurs (but also many other extinct reptiles and amphibians) is extraordinarily difficult unless the degrees of dimorphism are extreme. Prolonged growth, and the overlap of males and females in terms of body size and even features linked to sexually selected structures (such as the width of the terminal rosette seen here), make the sexual identity of individual specimens highly cryptic.

## ABBREVIATIONS

| | |
|---|---|
| **BSL** | Basal skull length (premaxilla to occipital condyle) |
| **IW** | Interorbital width |
| **MW** | Maximum width of skull (across quadratojugals) |
| **MWAO** | Maximum width across orbits |
| **NFML** | Narial fossa maximum length |
| **NFMW** | Narial fossa maximum width |
| **NML** | Naris maximum length |
| **NMW** | Naris maximum width |
| **OCW** | Occipital condyle width |
| **PBL** | Average pterygoid bulla length |
| **PEMW** | Premaxillary expansion maximum width |
| **SL_RTO** | Snout length rostral to orbits |
| **SMW** | Snout minimum width (mid-length) |

## ACKNOWLEDGEMENTS

We thank the following colleagues and curators for access to specimens, photographs, and computed tomographic data: Victoria Arbour, Salvator Bailon, Jordan Bestwick, Dave Blackburn, Ashley Burns-Meerschaert, Mark Carnall, Jim Clark, Tannis Davidson, Emily Durkin, Michael Franzen, Jennifer Gallichan, Dan Gordon, Ashely Hall, Natalie von Hoose, David Kizirian, Gunter Koehler, Jeff Lang, Josh Mata, Colin McHenry, Zachary Morris, Emma-Louise Nicholls, Olivier Pauwels, Stephanie Pierce, Rhian Rowson, Scott

Sampson, Mark Scherz, Coleman Sheehy III, Ed Stanley, Jeff Streicher, Frank Tillack, Zena Timmons, Paolo Viscardi, Kent Vliet, Lauren Vonnahme, Aki Watanabe, Gregory Watkins-Colwell, May Webber, Rom Whitaker, Nikhil Whitaker. Mathew Roloson kindly helped with data organization. We thank Peter Dodson, Michelle Stocker and Jeff Lang for their comments as referees, and John Hutchinson for his editorial comments, all of which helped improve earlier versions of this manuscript.

### Funding

This work was supported by the United States National Science Foundation Grants IBN-9601174, IBN-0343744, IOB-0517257, IOS-1050154 and IOS-1456503. The funders had no role in study design, data collection and analysis, decision to publish, or preparation of the manuscript.

### Grant Disclosures

The following grant information was disclosed by the authors:
United States National Science Foundation: IBN-9601174, IBN-0343744, IOB-0517257, IOS-1050154, and IOS-1456503.

### Competing Interests

The authors declare that they have no competing interests.

### Author Contributions

- David Hone conceived and designed the experiments, performed the experiments, prepared figures and/or tables, authored or reviewed drafts of the paper, and approved the final draft.
- Jordan C Mallon conceived and designed the experiments, performed the experiments, analyzed the data, prepared figures and/or tables, authored or reviewed drafts of the paper, and approved the final draft.
- Patrick Hennessey performed the experiments, authored or reviewed drafts of the paper, and approved the final draft.
- Lawrence M Witmer performed the experiments, prepared figures and/or tables, authored or reviewed drafts of the paper, and approved the final draft.

### Data Availability

  All the raw measurements, specimen numbers, and locations are available in Appendix 1.

### Supplemental Information

Supplemental information for this article can be found online at http://dx.doi.org/10.7717/peerj.9134#supplemental-information.

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
