# Peer review of "Ontogeny of a sexually selected structure in an extant archosaur Gavialis gangeticus (Pseudosuchia: Crocodylia) with implications for sexual dimorphism in dinosaurs"

_PeerJ, doi:10.7717/peerj.9134_

## Round 0.1 · original submission · Minor Revisions

Two reviewers agree that the MS is sound science and interesting, and publishable with moderate revisions. These reviews are constructive, which is much appreciated. A main theme is that they want the text tightened up to be shorter (fewer run-on) sentences and some more explanations of the literature, ideas or claims (e.g. dimorphism in archosaurs). A third review is much more critical, albeit with constructive points, and requires convincing that the analysis is robust and not overly repetitive of some past work. Please ensure all points are addressed individually in your Response. The MS will need some re-review. Thank you!

·

Basic reporting

This is an interesting study of growth and sexual dimorphism in Gavialis and I look forward to seeing it in print in due course. It is generally well written, although it is a bit sloppy at the start. The sample size and size range are both quite positive. It is surprising how poorly studied these crocodilians are. What is their conservation status? Are they endangered? Are there estimates of population size? The recognition of sexual dimorphism at all in crocodilians is quite recent. Who was the first to make such a case? It would be useful to summarize briefly, even in tabular form, the taxa that have been studied and the nature of the sexual dimorphism – e.g., size of structural? Apart from gharials, is there any other example within crocodilians of structure-based sexual dimorphism?

Experimental design

As far as the nuts and bolts of the study are concerned, I was a little surprised that the study approach was a nuts-and-bolts caliper-based approach. The more modern approach is landmark-based geometric morphometrics, which is well-suited to this study in particular, based as it is on analysis of photos. Nonetheless, I would be hard-pressed to state that the outcome of the study would be different if modern methods were used. As a classical bivariate study, I note the highly correlation among variables. Snout -vent length was reported but it was not selected as the standard variable on the grounds that measures of the snout were variable. With a R2 of 0.93, it is not drastically different from other snout variables, including SMW with R2 of 0.966. Allometric coefficients are readily interpreted with reference to overall skull length, less intuitively so with skull width. I recommend the former. In either case, I recommend that the male sample and the female sample be compared by means of a PCA and discriminant function analysis. These are fairly powerful and useful tools.

Validity of the findings

I am still a bit dubious about implications for detection of sexual dimorphism in the fossil record. How can it be that “no dinosaur has been determined to exhibit sexual dimorphism under rigorous analysis?” Is the fossil record really that lousy?! Are sample sizes too low? Is preservation too poor? In the case of sexual dimorphism by size alone, that seems an impossible barrier to cross. But in the case of sexual dimorphism by structure, as in Gavialis, it seems that enhanced sample size should not be required. How rare is this?

Additional comments

l. 44: “hereafter simply ‘dinosaurs’” Thank you so much – this brings tears to my eyes!!!
l. 45: “sexual size dimorphism” – I find this awkward grammatically. Would you consider “sexual dimorphism by size” or “size-based sexual dimorphism?” I hate to divide the phrase/concept “sexual dimorphism.
l. 50: tell us the taxa!
l. 56: elephants and humans have fairly modest growth rates
l. 58: you apply the word “rapidly” both to large mammals and small reptiles. This is sloppy and cries out for more precision.
l. 60: you are making an unexplained correlation between growth rate and sexual dimorphism. Surely sexual dimorphism occurs where it occurs and is often rampant in birds. I think you your problem is more practical than theoretical. Birds grow so fast it is hard to track the development of sexual dimorphism. The real advantage of crocodilians is that they grow slowly over a period of years and thus are excellent for any type of ontogenetic study.
l. 63: I hope you tell us more about sexual dimorphism in crocodilians. I could not have told you that they are often sexual dimorphic with exception of gharials.
l. 222: is “foraging” the best word? It often (but not always) connotes herbivory.
l. 235: bullae are NOT a visual signal, are they? “these features” implies that they are.
l. 250: bullae in alligators vastly more modest than those of gharials. I hesitate to call them similar.
l. 310: what about the possibility that an inflated bulla actually increases the surface area for origin of palatal muscles? It is not necessarily a negative.
l. 352: thank
l.376: andevolutionary
Fig. 2 the nasopharyngeal duct is a soft-tissue structure contained within the palatines. The figure incorrectly labels the palatines as the duct itself.
Fig. 5. The slope of the RMA plots is the allometric coefficient itself? So state and include slope on plot itself.
I recommend that tables S1 and S2 be included with the text rather than being in the supplemental files.

·

Basic reporting

Throughout the text the authors mainly have a consistent formatting style. I find it a little difficult to read in places because of multiple run-on sentences that contain too many thoughts and clauses. There are some issues with spelling and punctuation, and I corrected some of these. Please check the manuscript carefully.

The focus of this manuscript is extant taxa but aimed to collect predictive data for extinct, and as such the authors used morphological features that could be preserved in the fossil record. However, I found that the discussion focused more on what the structures could be for (rather than if there was a clear male/female distinction useful for fossils) and began to veer away from the main theme and (briefly) address ecological and dietary implications for the extant. Though these are important points to consider, I found that I wanted there to be more discussion and supporting data or references for several of the statements made (e.g. the resonance capabilities of the pterygopalatine bullae, the ability to deal with larger prey). Some things need to be addressed earlier in the text (the definition of what the bullae actually are), and that would make the paper flow a bit better.

Figures are extremely important for anatomical descriptions. In general, I found the figures here to be fairly informative. A comparative image of the pterygoid bullae of Gavialis and Alligator (because it was mentioned in the text) would be useful, as would comparative images of the largest and smallest individuals to see similarities and differences in the ghara/fossae across the size range sampled. The caption for Figure 1 is missing reference to image F- a small oversight. In general, the authors could have a few more call outs to the figures in the text.

Experimental design

The authors add to our knowledge of the ghara in Gavialis. However, I still wish that there was more concrete data for male/female of the specimens in order to avoid any circularity around size. The 2D measurement data are important variables for looking at shape change as related to size, and I think these could be complemented with some qualitative statements regarding the morphological differences among the ghara observed. I’m not certain that the authors actually tested sexual selection here, even though that is a prominent aspect of the title.

Validity of the findings

The authors discuss their results in terms of statistics, and I do not have the necessary background to comment on the appropriateness of the methods used. Their findings support what we already say: that the largest individuals are males, have ghara, and have enlarged bullae. The asynchronous appearance of skeletal maturity, ghara appearance, and bulla inflation appear to be new findings, or at least are more clearly documented here. I’m not sure that the appearance of these morphological features is as clearly tied to sexual selection (as in the authors’ title) as it could be (was this actually tested here?), and the ‘implications for sexual selection in dinosaurs’ (also in title) could be more clearly stated (right now, I’m interpreting the implications as ‘we still don’t have enough data to know in dinosaurs’- correct?).

Additional comments

This paper presents a new compilation of data on an interesting morphological structure. This paper needs some slight cleaning up throughout and more focus in the discussion. The quantification of these features through size (=ontogeny here) represents a step toward trying to interpret sexual dimorphism in the fossil record, a notoriously tricky thing. If the presence or exaggeration of these features may have ecological or trophic implications (which is reasonable), the authors need to add a bit more to frame those statements more clearly.

Reviewer 3 ·

Basic reporting

See attachment

Experimental design

See attachment

Validity of the findings

See attachment

Additional comments

See attachment

Annotated reviews are not available for download in order to protect the identity of reviewers who chose to remain anonymous.

---

## Round 0.2 · Minor Revisions

One reviewer has not responded (probably due to pandemic chaos), one has provided a helpful review, and one originally more critical reviewer has bowed out of the process. I feel comfortable proceeding because the prior review round's decision was minor revisions overall. The MS has improved.

The current reviewer makes some good suggestions. Allaying concerns about circularity by shifting all "male" specimen assignments in the MS to "presumed male" will help, and will reduce risk of future studies mistakenly re-using these data as 100% definitely male. I agree you should tone down the claims of sexual selection as what has been inferred here is sexual dimorphism (and with the aforementioned assumptions).

On the IUCN status of the gharial I suggest you cite this peer-reviewed paper: Lang, J, Chowfin, S. & Ross, J.P. 2019. Gavialis gangeticus (errata version published in 2019). The IUCN Red List of Threatened Species 2019: e.T8966A149227430. http://dx.doi.org/10.2305/IUCN.UK.2019-1.RLTS.T8966A149227430.en

The revised MS will simply be checked by me; not re-reviewed; and please do still include a Tracked Changes MS and clear Rebuttal. Thank you, and good health to all.

·

Basic reporting

The authors have made changes to make the text more readable, though I still maintain that the sentences are quite long. It is better though. The rearrangement of several sentences/short sections was helpful as well. I've fixed a few more grammatical and spelling issues that I found.

Figures are improved- thank you for including the comparative image of the alligator pterygoid bullae.

Experimental design

Though I am still concerned about circularity of data by assigning 'male' to the largest specimens without that actually recorded, the authors did add some text to qualify that this is an assumption. This could be made even more explicit by saying 'presumed male' rather than 'male' in all cases.

I also still agree with other reviews concerned with whether sexual selection was actually tested rather than just presence of sexual dimorphism. True, this is laid out as a hypothesis that can be tested with additional data in the future, but I'm still skeptical about it here.

Validity of the findings

The size of the dataset is emphasized repeatedly- yes, it is quite large and the authors are commended for accumulating those data. I still maintain that the findings are not entirely surprising. In the authors' response regarding dinosaurs in which a large enough sample exists to test sexual dimorphism, they bring up Coelophysis. However, this taxon was shown to exhibit a broad range of intraspecific variation in developmental patterns (Griffin 2018) rather than a bimodal dimorphic set of morphological features.

---

## Round 0.3 · accepted · Accept

The revisions follow what the reviewer and I recommended, and the MS has improved. Thank you for your efforts-- and congratulations!